# A Retrospective Analysis of Porcine Circovirus Type 3 in Samples Collected from 2008 to 2021 in Mexico

**DOI:** 10.3390/v15112225

**Published:** 2023-11-08

**Authors:** Mónica Reséndiz-Sandoval, Verónica A. Vázquez-García, Kenneth Contreras-Vega, Edgar A. Melgoza-González, Verónica Mata-Haro, Luis Gimenez-Lirola, Jesús Hernández

**Affiliations:** 1Laboratorio de Inmunología, Centro de Investigación en Alimentación y Desarrollo, A.C., Hermosillo 83304, Sonora, Mexico; mresendiz@ciad.mx (M.R.-S.); vavg990918@gmail.com (V.A.V.-G.); kennethcv4@gmail.com (K.C.-V.); edgaralonso.mglez@gmail.com (E.A.M.-G.); 2Laboratorio de Microbiología e Inmunología, Centro de Investigación en Alimentación y Desarrollo, A.C., Hermosillo 83304, Sonora, Mexico; vmata@ciad.mx; 3Department of Veterinary Diagnostic and Production Animal Medicine, College of Veterinary Medicine, Iowa State University, Ames, IA 50011, USA

**Keywords:** porcine circovirus 3 (PCV3), ELISA, real-time PCR, growth retardation, reproductive failure

## Abstract

Porcine circovirus type 3 (PCV3) is a nonenveloped virus of the *Circoviridae* family. This virus has been identified in pigs of different ages and pigs with several clinical manifestations of the disease or even in apparently healthy pigs. While PCV3 was first reported in 2015, several retrospective studies have reported the virus before that year. The earliest report indicates that PCV3 has been circulated in swine farms since 1996. In this study, we evaluated the presence of PCV3 in samples collected in Mexico in 2008, 2015, 2020, and 2021. This study assessed PCV3 DNA by qPCR and antibodies against CAP protein by indirect ELISA. The results showed that PCV3 (DNA and anti-CAP antibodies) was detected in the samples collected from 2008 to 2021. The highest prevalence was in 2008 (100%), and the lowest was in 2015 (negative). Genetic analysis of ORF2 showed that the virus identified belonged to genotype a, as most of the viruses identified thus far. PCV3 was detected in samples from piglets with respiratory signs and growth retardation, sows with reproductive failure, or asymptomatic piglets and sows. Pigs with respiratory signs, growth retardation, or reproductive failure had a higher prevalence of antibodies and qPCR-positive samples. In conclusion, this study showed that PCV3 has been circulating in Mexico since 2008 and that PCV3 DNA and antibodies were more prevalent in samples from pigs with clinical manifestations of diseases.

## 1. Introduction

Porcine circovirus includes four small single-stranded DNA viruses: PCV1, PCV2, PCV3, and PCV4 [1]. These nonenveloped viruses belong to the genus *Circovirus* and the *Circoviridae* family [1]. Currently, only PCV2 and PCV3 are considered pathogenic and responsible for health problems in the swine industry. PCV2 has been reported worldwide and is responsible for porcine circovirus diseases (PCVD) and porcine circovirus-associated diseases (PCVAD) in pigs [2]. PCV3 was first reported in the USA in 2015 in a farm with high mortality and low conception rates [3]. Since that report, the virus has been reported in many countries from Europe [4,5,6], Asia [7,8,9], and South America [10,11,12].

Similar to other porcine circoviruses, PCV3 contains three open reading frames (ORFs): ORF1, encoding the replicase protein; ORF2, encoding the structural protein or CAP protein; and ORF3, encoding a protein with an unknown function. ORF2 contains approximately 2000 nucleotides and has been used to evaluate the genetic characteristics of PCV3 and to compare several viruses identified worldwide [13]. Even with some discrepancies in its nomenclature, most of the studies support the existence of at least two genotypes (a and b) [14]. As in other porcine circoviruses, PCV3 CAP protein is the most antigenic and immunogenic protein of the virus and the target of the antibody response [15,16].

PCV3 DNA and antibodies against PCV3 have been detected in apparently healthy pigs and animals of several ages and with reproductive, respiratory, gastrointestinal, or neurological problems [8,17,18]. In sows, PCV3 has been detected in cases of mortality, reproductive failure, mummified fetuses, aborted fetuses, colostrum, stillborn, decreased neonatal rate, and porcine dermatitis nephropathy syndrome (PDNS) [19,20,21,22]. In piglets, PCV3 has been detected in animals with dyspnea, anorexia, fever, abdominal breathing, respiratory problems, diarrhea, wasting, neurological signs, and PDNS [20,21]. The virus has been detected alone or in coinfection with other pathogens, such as PCV2 [23], porcine reproductive and respiratory syndrome virus (PRRSV), porcine parvovirus, and classical swine fever [24]. However, several doubts remain about this virus, such as viral persistence; primary replication site(s); the factors involved in the clinical manifestations of positive pigs; and the clinical, pathological, and diagnostic characteristics to define PCV3 as responsible for a case [18,25].

Retrospective studies detected PCV3 before 2015, the first report of this virus. Sun and colleagues reported the presence of PCV3 in samples collected in China in 1996 [26]; similar results were reported in a study from Spain [16]. Ge and colleagues (2021) evaluated anti-PCV3 antibodies in a study from China; this study found a 42.87% prevalence. These antibodies were more frequent in sows (62.22%), followed by fattening pigs (28.96%), suckling pigs (8.96%), and nursey pigs (11.79%) [27]. Deng and colleagues (2018) observed similar prevalence percentages, where 52.6% prevalence was observed in serum samples from 2011 to 2017. In this case, the earliest positive samples were from 2012 [28]. Similarly, Geng and colleagues (2019) reported 22.35% and 51.88% prevalence for samples collected between 2015 and 2017, respectively [16]. Several reports have described PCV3 DNA in retrospective studies, but only a few have reported PCV3 DNA and anti-PCV3 antibodies. This complementary information provides additional information to better understand PCV3 evolution. This study aimed to evaluate the presence of PCV3 in samples collected in Mexico in 2008, 2015, 2020, and 2021.

## 2. Materials and Methods

### 2.1. Samples

Serum and tissue samples (minced and homogenized) were submitted to the Laboratorio de Inmunología, CIAD (Hermosillo, SON Mexico) for diagnostic purposes between 2008 and 2021. Samples collected in 2008 and 2021 corresponded to piglets with respiratory signs and growth retardation or sows with reproductive failure. Samples collected in 2015 and 2020 corresponded to asymptomatic piglets or sows. Based on the available clinical information, samples from piglets were classified as “growth retardation” or “asymptomatic”, and samples from sows were classified as “reproductive failure” or “asymptomatic”.

Samples included 52 serum samples from sows and 391 serum samples from piglets. For the analysis of anti-PCV3 antibodies, samples were evaluated individually, but for the detection of PCV3 DNA, samples were analyzed in pools (Table 1). A total of 89 samples from piglets from 2008 were organized into 12 pools. Samples from 2015 included 48 sera from piglets and 32 from sows; we prepared 6 pools from piglets and 5 pools from sows. Only 30 samples collected in 2020 were available and grouped into 4 pools. Most samples were collected in 2021 (*n* = 270) and organized into 52 pools. Pools consisted of 3, 6, or 8 samples, depending on the age and year of sampling. Tissue samples collected in 2021 included 5 minced and homogenized tissues from aborted fetuses. These samples were analyzed to detect PCV3 DNA by PCR only. Additionally, 50 negative serum samples from cesarean-derived and colostrum-deprived (CDCD) piglets were provided by Dr. Luis Gimenez-Lirola at Iowa State University (Ames, IA, USA). These samples were negative for PCV3 DNA and had no anti-PCV3 antibodies. CDCD samples were collected from a previous study performed by Dr. Gimenez-Lirola’s research group (Veterinary Diagnostic and Production Animal Medicine, College of Veterinary Medicine, Ames, IA, USA). This study was approved by AMVC WeSearch DBA VRI (Audubon, IA, USA) Animal Use and Care Committee (BI-S-18-1248).

### 2.2. Real-Time PCR

The DNA from sera and tissues was extracted using a QIAamp DNA Mini kit (Qiagen, Hilden, Germany) following the manufacturer’s recommendations. Sera were pooled according to age, farm, and year. Tissues were analyzed individually. Real-time PCR (qPCR) analysis to detect ORF2 was performed in a final volume of 25 μL containing 150 nM primers (forward TGTWCGGGCACACAGCCATA and reverse TTTCCGCATAAGGGTCGTCTT) [29], 10 μL of Brilliant III Ultra-Fast SYBR Green qPCR Master Mix (Agilent Technologies, Santa Clara, CA, USA), 10.25 μL of water, and 4 μL of DNA. PCR amplification was performed in a StepOne Real-Time PCR system (Applied Biosystem, Foster City, CA, USA) as follows: 35 cycles at 95 °C for 3 min, 94 °C for 5 s, 60 °C for 10 s, and 72 °C for 5 s. The Ct was defined by adjusting the threshold in the exponential phase, and we used a positive control to confirm that the threshold was the same in all experiments. Samples with Ct values > 35 were considered negative.

Samples with Ct values < 33 were further processed for ORF2 sequencing. Conventional PCR was performed to amplify the *ORF2* gene of PCV3 in a final volume of 20 μL containing 300 nM primers (forward TTACTTAGAGAACGGACTTGTAAC and reverse AAATGAGACACAGAGCTATATTCAG) [18], 12.5 μL of KAPA HiFi HotStart ReadyMix, 6 μL of water, and 5 μL of DNA. PCR amplification was performed in an MJ Mini Personal Thermal Cycler (Bio-Rad, Hercules, CA, USA) as follows: 35 cycles at 95 °C for 3 min, 94 °C for 20 s, 62 °C for 15 s, 72 °C for 60 s, and 72 °C for 8 min. Successful amplification was determined when a PCR product of 660 bp was obtained and confirmed by 1.2% agarose electrophoresis.

### 2.3. PCV3 Sequencing and Phylogenetic Analysis

The phylogenetic analysis of 8 ORF2 PCV3 samples was aligned with nucleotide sequences obtained from GenBank, and we built a tree with the sequences of PCV3 isolated from different parts of the world. Multiple sequence alignment and sequence comparisons were made in DNAstar Lasergene software version 17.3 (Madison, WI, USA) by 1000 repetitions of Randomized Accelerated Maximum Likelihood (RAxML). A phylogenetic tree was prepared with the Interactive Tree of Life version 6 (iTOL) [30].

### 2.4. ELISA

We used a PCV3 CAP protein produced by Dr. Luis Gimenez-Lirola [21]. Maxisorp ELISA microwell plates (Nunc, Thermo Fisher Scientific, Waltham, MA, USA) were used to coat 2 µg/mL PCV3 CAP protein diluted in coating buffer (ImmunoChemisty Technologies, Davis, CA, USA) and incubated overnight at room temperature (≈25 °C; all incubations were performed at room temperature). To block the ELISA plates, we used a General Block buffer (ImmunoChemisty Technologies, Davis, CA, USA) by incubating the plates for 24 h. Then, plates were kept at 4 °C until use. Serum samples were diluted 1:100 using General Sample Diluent (ImmunoChemisty Technologies, Davis, CA, USA) and incubated for 30 min with slow agitation. Then, the wells were washed five times with PBS with 0.1% Tween 20 (PBST), and 50 µL of goat anti-porcine IgG-HRP (Polyclonal; Cat. No. 6050-05) was added to the plate and incubated for 30 min with slow agitation. After five washes with PBST, 50 µL of 3,3’,5,5’-tetramethylbenzidine (ImmunoChemistry, Davis, CA, USA) was added for 20 min. The reaction was stopped with 50 µL of 1 M H_2_SO_4_, and the optical density (O.D.) was read at 450 nm using an automated spectrophotometer (Thermo Scientific Multiskan FC Microplate Photometer, Waltham, MA, USA). Each plate included a positive control (pool of sera from PCV3-infected swine) and a negative control (pool of sera from colostrum-deprived pigs) in duplicate and blanks. The mean of the blanks was subtracted from the absorbance of the samples, and the results were expressed as the O.D. To define the cutoff, we used ROC curve analysis of the O.D. of sera from PCV3-infected swine and the sera from colostrum-deprived pigs.

### 2.5. Statistical Analysis

The cutoff for the indirect ELISA was determined using ROC curves, as well as the diagnostic sensitivity and specificity and AUC. The differences in PCV3 IgG antibodies between years were analyzed by the Kruskal—Wallis test and multiple comparisons with Dunn’s test. The differences in PCV3 IgG antibodies between sows with reproductive failure and asymptomatic sows and the differences between piglets with growth retardation and asymptomatic piglets were analyzed by the Mann—Whitney test. In all cases, the analyses were performed with a significance level of 0.05 in the statistical analysis package GraphPad PRISM version 8.0.2.

## 3. Results

### 3.1. Detection of PCV3 DNA in Serum and Tissue Samples

PCV3 DNA was analyzed in serum samples collected between 2008 and 2021. In the case of minced and homogenate samples from aborted fetuses (tissue samples), only samples collected in 2021 were available. Table 2 shows that PCV3 was detected in 4 of 5 tissue samples with a Ct value ranging from 22 to 32.

In the case of serum samples, the presence of PCV3 was evaluated in pools of samples organized by year, as described in the Section 2 and Table 1. All pools collected in 2008 were positive, with Ct values ranging from 28.18 to 33.37. The analysis of samples collected in 2015 showed that 5 of 11 pools were positive, with Ct values ranging from 30.14 to 33.54. Positive samples from 2015 corresponded to piglets, and all pools from sows were negative (*n* = 5). No positive pools were detected in samples collected in 2020, and only 3 of 52 pools from 2021 were positive (Ct values ranging from 29.11 to 34.7) (Table 3).

### 3.2. Genetic Analysis

To analyze the genetic characteristics of PCV3 detected in this study, samples with a Ct value < 33 were used to amplify the *ORF2* gene and sequenced. The *ORF2* gene was successfully sequenced from eight samples: four from serum samples collected in 2008 and four from minced and homogenate samples from aborted fetuses collected in 2021.

A phylogenetic tree was constructed using 147 sequences obtained from GenBank and identified as genotypes “a” (*n* = 146) and “b” (*n* = 1), and the eight sequences obtained in this study (GenBank accession No. OR757332–OR757339) (Figure 1). The results showed that all eight sequences identified in this study corresponded to genotype “a”. Additionally, the analysis revealed that the sequences identified belong to a unique branch closely related to a sequence reported in China in 2015. The tree also showed many sequences closely related to the sequences reported in this study. Interestingly, this cluster includes many sequences from China and Latin America and two sequences from Mexico from 2012 and 2015. The sequences showed high identities between them (98–99%).

### 3.3. Detection of Anti-PCV3 IgG Antibodies in Serum Samples

To determine the presence of PCV3 CAP protein IgG antibodies, we established the experimental conditions of an indirect ELISA. As negative controls, we used serum samples from colostrum-deprived piglets. As positive samples, we used PCR-positive samples collected in 2008. Figure 2 shows the ROC curve (Figure 2a), revealing that with a cutoff of <0.4170, the assay had 100% diagnostic sensitivity and 100% diagnostic specificity (Figure 2b).

Figure 3 shows the prevalence of PCV3 IgG antibodies in all samples analyzed in this study (*n* = 443). The results showed that 43.34% (192 of 443) of the samples were seropositive for PCV3 (Figure 3a). When the analysis was performed by year (Figure 3b), in 2008, 100% (125 of 125) were seropositive; in 2015, all samples were negative (0 of 58); in 2020, 22.22% (6 of 27) were seropositive; and in 2021, 41.20% (96 of 233) were seropositive.

Figure 4 compares the PCV3 IgG antibodies in sows with reproductive failure versus asymptomatic sows (Figure 4a) and piglets with growth retardation versus asymptomatic piglets (Figure 4b). In the case of asymptomatic sows, all samples were seronegative (0 of 36). In contrast, 93.75% (15 of 16) of the sows with reproductive failure were seropositive. A similar scenario was observed in the case of samples from piglets. Only 5.20% (5 of 96) of the asymptomatic piglets were seropositive for PCV3, and 58.30% (172 of 295) of the piglets with growth retardation showed PCV3 antibodies.

## 4. Discussion

This study evaluated the presence of PCV3 in serum and tissue samples collected in Mexico in 2008, 2015, 2020, and 2021. PCV3 was assessed by qPCR to detect viral DNA and by indirect ELISA to detect anti-CAP protein IgG antibodies. Our results confirm that PCV3 circulated in pigs before 2015, the year of the first report of this virus in the USA. The earliest detection of PCV3 was in samples collected in 1996 in a study performed in China [26] and in a similar study from Spain [9]. Other reports from Thailand [31] and Brazil [10] described the presence of PCV3 in samples collected in 2006. However, previous studies did not report the prevalence by year, with only a prevalence of 36.7% and 47.8% for Thailand and Brazil, respectively. In our study, the oldest samples evaluated were collected in 2008. Contrary to other studies, our study indicates that all the samples collected in 2008 were positive for PCV3 (89 samples, 100% prevalence). Interestingly, the prevalence of PCV3 in the following years was lower. The overall prevalence of PCV3 DNA in the samples analyzed was 22% (17 of 77 pools). It is difficult to explain why, in 2008, the prevalence was higher, but we can hypothesize that the first infections of PCV3 could have occurred in that year (and perhaps previous years not included in this investigation), and the virus can disseminate easily in a naïve population. However, this hypothesis is difficult to test.

The diversity of PCV3 is lower than that of other Circoviruses, such as PCV2. Nevertheless, some authors have suggested the presence of several genotypes [32,33,34]. Meanwhile, others have proposed only two genotypes, a and b [14]. We agree that PCV3 can currently be classified into genotypes a and b. In this line, the eight sequences obtained in this study and 147 sequences obtained from the literature were used to construct a phylogenetic tree. The analysis revealed that sequences obtained in this study belong to genotype “a” but are organized in a single branch but with a close relation to a sequence reported in China in 2015. Our phylogenetic analysis also revealed two main clusters; interestingly, the cluster where our sequences are grouped also includes many sequences from America and two sequences from Mexico previously deposited in GenBank. The other cluster also contains sequences from America but in a lower number. In further studies, it will be interesting to perform a deep analysis of all the sequences described from America to study the evolution of PCV3 in America. Additionally, obtaining more sequences from recent years is necessary to obtain a more detailed picture of PCV3 evolution in Mexico.

The seroprevalence of PCV3 has been poorly evaluated [16,27,28]. In this study, a PCV3 indirect ELISA based on the CAP protein was established and used to evaluate the prevalence of PCV3 antibodies in Mexico. Our results showed an overall prevalence of 43.34%, but the analysis by year revealed that the 2008 and 2021 samples had a higher prevalence of anti-PCV3 antibodies than samples collected in 2015 and 2020 (*p* < 0.05). No positive cases were detected in samples from 2015, and only a few positives were observed in sera from 2020 (Figure 3a). The samples used in this study were not collected with the aim of evaluating the prevalence of PCV3 or another pathogen, which allowed us to say that the negative status of samples from 2015 is not indicative that PCV3 did not circulate in that year, only that these samples were negative for PCV3. Samples from 2015 corresponded to asymptomatic animals that were negative for PCV3 IgG antibodies, but 5 of 12 pools were PCR positive, suggesting an early infection and confirming the circulation of PCV3 in 2015. A similar explanation applies to samples from 2020, where all pools were negative (*n* = 4), with 22% seropositivity to PCV3 IgG antibodies. In the case of samples from 2021, high seropositivity with a low presence of viral DNA confirmed the circulation of PCV3 in Mexico.

To support this statement, we analyzed the samples according to their health status. An asymptomatic status implies that samples belong to pigs without clinical manifestations of diseases. In contrast, the group of sows with reproductive failure or piglets with respiratory disease and growth retardation. The antibody analysis according to this classification showed that the samples from sows and pigs in the asymptomatic group had a low prevalence of antibodies, contrary to the animals with growth retardation or reproductive failure, where the prevalence was high. These results agree with previous reports, indicating that some asymptomatic pigs had PCV3 antibodies and that most pigs with growth retardation or reproductive failure had PCV3 antibodies. However, it is important to note that this study was not aimed at associating PCV3 antibodies with an infection caused by PCV3. Even so, the analysis of PCV3 antibodies and PCV3 DNA revealed that asymptomatic pigs are negative for antibodies but could be positive for PCR, and pigs with growth retardation or reproductive failure are positive for antibodies and positive by PCR. These results support the idea that PCV3 is a ubiquitous pathogen and can be detected in different health statuses, asymptomatic, pigs with growth retardation or reproductive failure; however, pigs with clinical manifestations of the disease can also produce anti-PCV3 antibodies and higher viral loads. This hypothesis requires further studies to determine the causes provoking ubiquitous clinical manifestations.

## 5. Conclusions

This retrospective study aimed to evaluate the prevalence of PCV3 in samples collected from several regions of Mexico between 2008 and 2021. This study identified the presence of PCV3 (DNA and IgG antibodies) in samples collected as early as 2008. The genetic analysis revealed that similar to other PCV3 viruses, the ORF2 sequences obtained belong to genotype a. The analysis of antibodies showed different seroprevalence by year, while the classification, according to the health status, showed a low seroprevalence in asymptomatic pigs, contrary to pigs with growth retardation or reproductive failure. In summary, this study confirms that PCV3 circulates in several countries before the first report in 2015 and that most pigs with clinical manifestations of the disease have PCV3 antibodies.

## Figures and Tables

**Figure 1 viruses-15-02225-f001:**
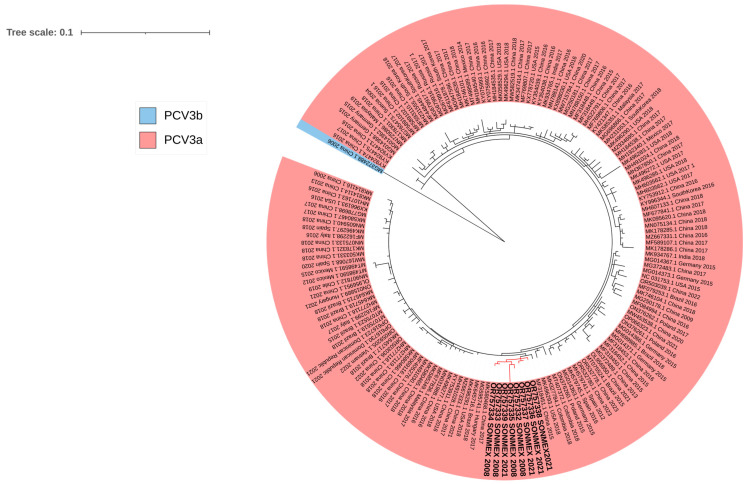
The phylogenetic tree using ORF2. The tree was obtained using multiple sequence alignment, and sequence comparisons were made in DNAstar Lasergene software version 17.3 and by the Randomized Accelerated Maximum Likelihood (RAxML) 1000 repetitions. Interactive Tree of Life (iTOL) was used for tree visualization. The scale bar indicates nucleotide substitutions per site. The sequences obtained in this study are highlighted in bold.

**Figure 2 viruses-15-02225-f002:**
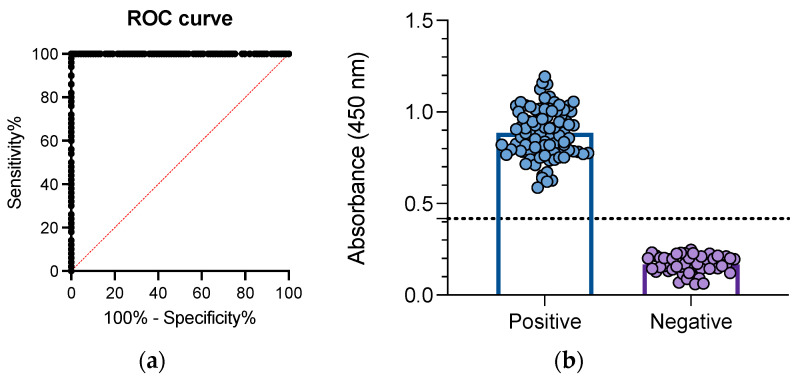
Standardization of an indirect ELISA to detect IgG antibodies against PCV3. PCV3-positive samples were used as a positive control. Negative samples consisted of serum from colostrum-deprived (CD) piglets. The ROC curve (**a**) was constructed and used to set the cutoff at 0.4170 (**b**).

**Figure 3 viruses-15-02225-f003:**
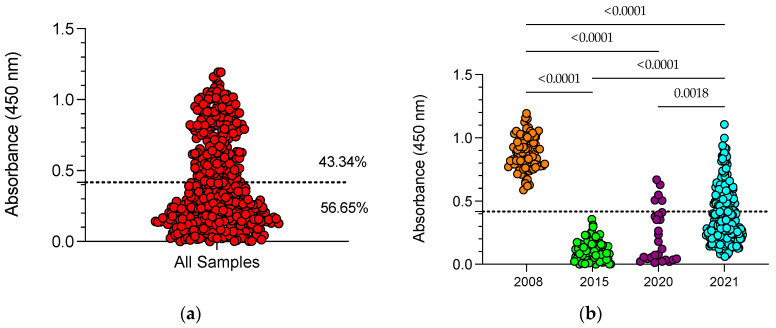
Prevalence of PCV3 antibodies. PCV3 IgG antibodies were evaluated in all the samples ((**a**), *n* = 443 samples) or in the same samples, but according to the year of sampling ((**b**), 2008 *n* = 125; 2015 *n* = 58; 2020 *n* = 27; 2022 *n* = 233). Significant differences between PCV3 IgG antibodies between years were analyzed by the Kruskal—Wallis test and multiple comparisons with Dunn’s test.

**Figure 4 viruses-15-02225-f004:**
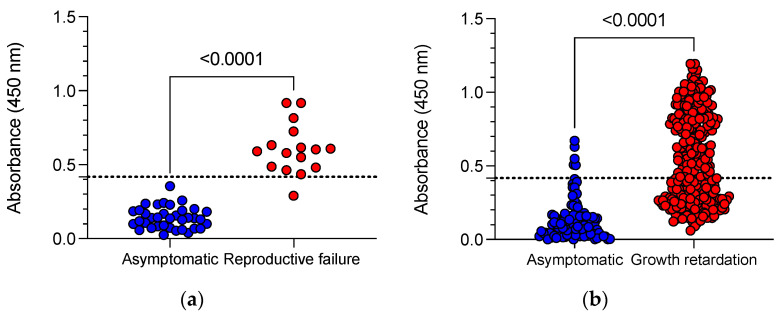
Distribution of PCV3 IgG antibody levels according to health status. Samples were grouped according to the clinical history of the animals (growth retardation or reproductive failure, or asymptomatic) and divided into samples from sows (**a**) or piglets (**b**). Significant differences between PCV3 IgG antibodies between years were analyzed with the Mann—Whitney test.

**Table 1 viruses-15-02225-t001:** Samples analyzed by qPCR.

Year	Number of Samples			Number of Pools		
	Piglets	Sows	Tissues *	Piglets	Sows	Tissues
2008	89	0	0	12	0	0
2015	48	32	0	6	5	0
2020	30	0	0	4	0	0
2021	270	0	5	52	0	0

* Tissue consisted of minced and homogenized samples. In this case, no pools were prepared, and samples were analyzed individually only by qPCR.

**Table 2 viruses-15-02225-t002:** PCV3 detection in minced and homogenized samples ^1^.

Year	Sample ID	GenBank Accession	Ct Value *
2021	10-P	-	28.00
2021	11-P	OR757337	22.00
2021	13-P	OR757338	26.00
2021	184-P	OR757336	32.00
2021	12-P	-	Negative

^1^ Minced and homogenate samples obtained from aborted fetuses. * Ct values > 35 were considered negative.

**Table 3 viruses-15-02225-t003:** Detection of PCV3 in serum samples.

Year	Pool Name *	IgG Anti-PCV3	Health Condition	Ct Value	GenBank Accession
2008	C1P1	Yes	Growth retardation	31.37	-
	C1P2	Yes	Growth retardation	30.96	-
	C1P3	Yes	Growth retardation	28.54	-
	C1P4	Yes	Growth retardation	30.0	OR757332
	C1P5	Yes	Growth retardation	28.18	OR757333
	C2P1	Yes	Growth retardation	29.06	-
	C2P2	Yes	Growth retardation	29.36	OR757334
	C2P3	Yes	Growth retardation	29.07	-
	C2P4	Yes	Growth retardation	29.17	OR757335
	C2P5	Yes	Growth retardation	30.89	-
	C2P6	Yes	Growth retardation	32.24	-
	C3P1	Yes	Growth retardation	33.37	-
2015	C4P1	No	Asymptomatic	32.21	-
	C4P2	No	Asymptomatic	33.54	-
	C4P3	No	Asymptomatic	31.95	-
	C4P4	No	Asymptomatic	31.5	-
	C4P6	No	Asymptomatic	30.14	-
2021	P2	Yes	Growth retardation	33.06	-
	P5	Yes	Growth retardation	29.11	OR757339
	P6	Yes	Growth retardation	34.7	-

* Only pools with Ct values ≤ 35 were included in this table. Pools with Ct values > 35 were considered negative (negative pools by year: 2008, *n* = 0; 2015, *n* = 7; 2020, *n* = 4; 2021, *n* = 49).

## Data Availability

PCV3 ORF2 sequences analyzed in our study are available at the NCBI GenBank with the accession numbers OR757332–OR757339.

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
