# Peer review of "A Retrospective Analysis of Porcine Circovirus Type 3 in Samples Collected from 2008 to 2021 in Mexico"

_viruses, 2023, doi:10.3390/v15112225_

Round 1

Reviewer 1 Report

Comments and Suggestions for Authors

In this manuscript titled “Retrospective analysis of PCV3 in samples collected from 2008 2 to 2021 in Mexico”, authors analyzed the presence of PCV3 in samples collected in Mexico in 2008, 2015, 2020, and 2021. The results confirmed that PCV3 has been circulating in Mexico since 2008 and that the genome and antibodies were more prevalent in samples from pigs with clinical manifestations of diseases. However, there exist many minor problems in this manuscript, which need further revision and improvement. The specific amendments are as follows:

1.      In line 18, “PCV3” is misspelled.

2.      In line 23, “ORF-2” should be written as “ORF2”.

3.      In line 34, “Circoviridae” should be in italics.

4.      In line 67, lack of punctuation after “Cols”.

5.      There are some formats that need to be noted: the “ORF2” should be in italics, in lines 111 and 166.

6.      In line 137, the “C” in “Immunochemistry” should be in uppercase format.

7.      In line 170, the punctuation in “b.” should be outside the colon, and the same applies to “a.” in line 171.

8.      In line 226, [14] should not use italics.

9.      In lines 170 and 171, “a” and “b” should be in italics.

10.  In lines 24, 226, 227, and 264, “a” should be in italics.

11.  Why are the sample analysis methods for anti-PCV3 antibodies and the PCV3 genome different?

12.  In Table 2, “anti PCV3” should be written as “anti-PCV3”. 

Author Response

Reviewer 1

In this manuscript titled “Retrospective analysis of PCV3 in samples collected from 2008 2 to 2021 in Mexico”, authors analyzed the presence of PCV3 in samples collected in Mexico in 2008, 2015, 2020, and 2021. The results confirmed that PCV3 has been circulating in Mexico since 2008 and that the genome and antibodies were more prevalent in samples from pigs with clinical manifestations of diseases. However, there exist many minor problems in this manuscript, which need further revision and improvement. The specific amendments are as follows:

 Thank you for your comments.

  1. In line 18, “PCV3” is misspelled.

Answer: Corrected as suggested.

  1. In line 23, “ORF-2” should be written as “ORF2”.

Answer: Corrected as suggested.

  1. In line 34, “Circoviridae” should be in italics.

Answer: Corrected as suggested.

  1. In line 67, lack of punctuation after “Cols”.

Answer: Corrected as suggested.

  1. There are some formats that need to be noted: the “ORF2” should be in italics, in lines 111 and 166.

Answer: Corrected as suggested.

  1. In line 137, the “C” in “Immunochemistry” should be in uppercase format.

Answer: Corrected as suggested.

  1. In line 170, the punctuation in “b.” should be outside the colon, and the same applies to “a.” in line 171.

Answer: Corrected as suggested.

  1. In line 226, [14] should not use italics.

Answer: Corrected as suggested.

  1. In lines 170 and 171, “a” and “b” should be in italics.

Answer: Corrected as suggested.

  1. In lines 24, 226, 227, and 264, “a” should be in italics.

Answer: Corrected as suggested.

  1. Why are the sample analysis methods for anti-PCV3 antibodies and the PCV3 genome different?

 Answer: Because several reports have confirmed that genome detection can be achieved in pools without affect the sensitivity, which in turn increase the number of samples (animals) that can be tested simultaneously and cost effectively. Contrary, the antibody detection is better performed indivually to avoid compromising the sensitivity of the test. PMID: 29435356   

  1. In Table 2, “anti PCV3” should be written as “anti-PCV3”. 

Answer: Corrected as suggested.

Reviewer 2 Report

Comments and Suggestions for Authors

Re-analysing the stored samples in a retrospective study to reveal the past prevalence of PCV3 is the main and unquestionable value of the  manuscript. Even the high number of analysed samples is impressive. Therefore publication of these data is important and puts a significant piece in the puzzle of the PCV3 story. Nevertheless the recent form of data analysis should be thought over to draw reasonable conclusions and publish appropriate data on this important field.

The fundamental weakness of the data analysis is the cutoff setting in ELISA method validation. Let me detail my concerns in points:

1. First of all "negative" sera are from CDCD piglets, who are not just negative, but naïve. They represents an extreme and uncommon population letting you to set the cutoff unrealistically low. What about, for example, sera samples from normal commercial piglets from a herd which is PCV3-free since generations, but vaccinated against and maybe contaminated by other pathogens? They would be really PCV3 negative, with a probably higher abs value. If you don't include such type of negatives in your analysis, you cannot demonstrate the width of abs values in natural negatives.

2. Moreover the ELISA data analysis is an example of circular reasoning, which is a definitive logical mistake. The dotplots in Figure 2b are exactly the same that in Fig 3a (negative) and in Fig3b (higher values in '2008'). So you had defined samples as positive and negative a' priori, validated your method by them, and finally reported the same samples are positive/negative. It seems arbitrary and unfounded. 

3. For cutoff setting you used PCR-positive samples collected in 2008 as positive controls (lines 181-182). But serum pools were analysed by qPCR while single samples were used in ELISA, aren't they? Was the pool results projected to all of the members? Is it correct? Moreover what about pools C4P1 and C4P2? According to your own definition they are PCV3 positive by qPCR, so they are 'positive', but you could't detect IgG (Table 2). I miss these datapoint from Fig 2b,  they were omitted intentionally. Of course, your ROC curve is more than perfect, with 100% sensitivity and specificity if you had said what is positive and what is negative. This is incorrect.

4. If your Fig 2 was accepted in recent form (No, it wasn't!), there is not explanation why 0.3 was set as cutoff? Why not 0.5? Shouldn't you set a 'grey' zone between positives and negatives?

To sum it up, you should revalidate the cutoff value with correct positive and negative samples, consider setting a 'grey' inconclusive zone. Thereafter recalculate the resulted prevalence% and upgrade the discussion.

There are some other major points to be corrected:

The definition of 'healthy' and 'unhealthy' animals are also tendentious. How on earth can the healthy animals be defined as 'nonreproductive failure or mortality' (line 84)? You should group the data objects according to symptome localization and/or severity, but not as simple as healthy/unhealthy.

Since your only numeric result from qPCR is Ct, e.g. not copy number, you should clearly write the threshold definition to make your analyisis repeatable.

Why did you used this particular phylogenetic analysis method? Could you involve any method considering timescale? 

Regarding to the tree: too big for printing in one page. To make it more informative enlarge the font, highlight the seqs from this study by color etc. and if all of your seqs are in group 'a', compress the group 'b', like a triangle.

Some minor comments:

line 21: CAP protein or CAP antigen

Using 'PCV3 genome' can be misleading when you refer not to genome sequence data but detection PCV3 DNA (nucleic acid) as an antigen. Change it through the m.s.

Rethink the keywords! Some of them are redundant, and no reference to qPCR.

line 75-77: This sentence in not part of introduction, delete!

line 81: CIAD (Hermosillo, MX)

in footnotes of Table 1 and 2: Insert number of negative samples, like this: 'Ct values >35 (n=...) were considered negative'!

Finally I have two questions for more discussion:

How can you explain the descending prevalence in time? 

According to phylogenetic analysis can you estimates anything about the origin of the Mexican PCV3 strains?

Author Response

Reviewer 2

Comments and Suggestions for Authors

Re-analysing the stored samples in a retrospective study to reveal the past prevalence of PCV3 is the main and unquestionable value of the  manuscript. Even the high number of analysed samples is impressive. Therefore publication of these data is important and puts a significant piece in the puzzle of the PCV3 story. Nevertheless the recent form of data analysis should be thought over to draw reasonable conclusions and publish appropriate data on this important field.

Thank you for your comments.

The fundamental weakness of the data analysis is the cutoff setting in ELISA method validation. Let me detail my concerns in points:

  1. First of all "negative" sera are from CDCD piglets, who are not just negative, but naïve. They represents an extreme and uncommon population letting you to set the cutoff unrealistically low. What about, for example, sera samples from normal commercial piglets from a herd which is PCV3-free since generations, but vaccinated against and maybe contaminated by other pathogens? They would be really PCV3 negative, with a probably higher abs value. If you don't include such type of negatives in your analysis, you cannot demonstrate the width of abs values in natural negatives.

Answer: We respectfully disagree that the use of this kind of sera is unrealistic. The main rational for using CDCD pigs to set up the cutoff is that was that these pigs are raised negative to most swine pathogens. Freedom of the pig for the pathogen of interest (i.e., PCV3) and other common swine pathogens (porcine circovirus type 2, porcine reproductive and respiratory syndrome virus, influenza A virus, mycoplasma sp) is important at study start. Moreover, using colostrum-deprived (CD) pigs is a good alternative strategy to circumvent passively acquired immunity against PCV3 or exposure to any circovirus that may occur shortly after birth. This is particularly important when studying pathogens of unknown but presumably high prevalence. However, too solve reviewer’s concern, we have updated the cutoff up to 0.4170. This value was defined by the ROC curve to provide a 100% diagnostic sensibility and specificity. This value also represents the mean of the CDCD sera plus five standard deviation. We think that with this cutoff our conclusions are better supported. Also, it is important to note that for this kind of infections, the use of “negative normal serum” is not easy. Several reports have confirmed a wide distribution of this virus around the World.      

  1. Moreover the ELISA data analysis is an example of circular reasoning, which is a definitive logical mistake. The dotplots in Figure 2b are exactly the same that in Fig 3a (negative) and in Fig3b (higher values in '2008'). So you had defined samples as positive and negative a' priori, validated your method by them, and finally reported the same samples are positive/negative. It seems arbitrary and unfounded. 

Answer: The negative samples were defined a’ priori, as the Reviewer explains. Because of the characteristics of the CDCD sera previously mentioned. Figure 3b was corrected and only control samples were included. In the case of the positive samples, we think that our best candidate for positive control could be the PCR positive samples, because it was the only control available. The result of PCR was known a’ priori, because these experiments were performed before running the ELISAs. The use of this controls was not probably the ideal option, yet it was the best option for us to perform our experiment. Even with this limitation, we think that the results are not affected, the conclusions are not arbitrary or unfunded.

  1. For cutoff setting you used PCR-positive samples collected in 2008 as positive controls (lines 181-182). But serum pools were analysed by qPCR while single samples were used in ELISA, aren't they? Was the pool results projected to all of the members? Is it correct? Moreover what about pools C4P1 and C4P2? According to your own definition they are PCV3 positive by qPCR, so they are 'positive', but you could't detect IgG (Table 2). I miss these datapoint from Fig 2b,  they were omitted intentionally. Of course, your ROC curve is more than perfect, with 100% sensitivity and specificity if you had said what is positive and what is negative. This is incorrect.

Answer: Samples CAP1 and CAP2 correspond to 2015, we have realized that we made a mistake in the line dividing the samples/year. Reviewer is right, PCR samples were analyzed in pool and individually by ELISA. After test all the samples collected in 2008, we realized that all were also positive to PCV3.

  1. If your Fig 2 was accepted in recent form (No, it wasn't!), there is not explanation why 0.3 was set as cutoff? Why not 0.5? Shouldn't you set a 'grey' zone between positives and negatives?

Answer: We have updated the cutoff to 0.4170, according to the ROC curve. We set the cutoff in .3 considering an intermediate value between 0.2315 and 0.4170, according to the ROC curve.  

To sum it up, you should revalidate the cutoff value with correct positive and negative samples, consider setting a 'grey' inconclusive zone. Thereafter recalculate the resulted prevalence% and upgrade the discussion.

Answer: As previously explained, the cutoff have been modified, from 0.3 to 0.4170 according to the ROC curve analysis, which provide a diagnostic specificity of 100%. We also modified the results and discussion accordingly. We think that at this moment is difficult to define a “grey” zone and we rather prefer to use a conservative cutoff.

There are some other major points to be corrected:

The definition of 'healthy' and 'unhealthy' animals are also tendentious. How on earth can the healthy animals be defined as 'nonreproductive failure or mortality' (line 84)? You should group the data objects according to symptome localization and/or severity, but not as simple as healthy/unhealthy.

Answer: We have modified the nomenclature to describe the groups, no, samples are defined as follow: samples of piglets were classified as “growth retardation” or “asymptomatic” and sows as “reproductive failure” or “asymptomatic”.

Since your only numeric result from qPCR is Ct, e.g. not copy number, you should clearly write the threshold definition to make your analyisis repeatable.

Answer: The Ct was defined by adjusting the threshold in the exponential phase and using a positive control to confirm the threshold was the same in all the experiments.

Why did you used this particular phylogenetic analysis method? Could you involve any method considering timescale?

Answer: The program we have to perform this analysis has two options: Maximum Likelihood (RAxML) or Neighbor-Joining (BIONJ). We chose RAxML because, in addition to being a popular program used in phylogenetic analysis, including PCV3, it uses the maximum likelihood.  

Regarding to the tree: too big for printing in one page. To make it more informative enlarge the font, highlight the seqs from this study by color etc. and if all of your seqs are in group 'a', compress the group 'b', like a triangle.

Answer: The tree has been improved as suggested.

Some minor comments:

line 21: CAP protein or CAP antigen

Answer: CAP protein. The change was done accordingly.

Using 'PCV3 genome' can be misleading when you refer not to genome sequence data but detection PCV3 DNA (nucleic acid) as an antigen. Change it through the m.s.

Answer: Thank you for the comment. We have changed “PCV3 genome” by “PCV3 DNA” through the m.s.

Rethink the keywords! Some of them are redundant, and no reference to qPCR.

Answer: We have updated the keywords accordingly.

line 75-77: This sentence in not part of introduction, delete!

Answer: Done as suggested.

line 81: CIAD (Hermosillo, MX)

Answer: Done as suggested.

in footnotes of Table 1 and 2: Insert number of negative samples, like this: 'Ct values >35 (n=...) were considered negative'!

Answer: Done as suggested in the table 2, in table 1 the information is already in the table.

Finally I have two questions for more discussion:

How can you explain the descending prevalence in time? 

Answer: Unfortunately, it is not possible to explain the descending prevalence in time.

According to phylogenetic analysis can you estimates anything about the origin of the Mexican PCV3 strains?

Answer: We have added a brief discussion about the possible origin of the Mexican PCV3 strains.

Reviewer 3 Report

Comments and Suggestions for Authors

Major comments

Line 61: please, add a suitable reference at this point.

Line 75: please, correct 2000 (maybe, 2020).

Lines 80-86: please, add more details about tissue samples, as well as about healthy samples. Moreover, please add details about ethical approval of the present study.

Lines 87-94: please, report such data as a table.

Lines 95-98: please, add more details about such samples. How was tested/detected the absence of PCV3 genome and anti-PCV3 antibodies? Who is Dr. Gimenez-Lirola? Please, add details about his affiliation. Were such samples from an ethically approved investigation?

Line 150: please, add such data within the materials and methods section.

Lines 160-161: actually, four pools 2015 are reported in table n. 2. Please, check such results.

Lines 162-163: two pools 2021 are reported in table n. 2. Please, check such results.

Lines 179-185: please, move such data to the materials and methods section.

Lines 219-220: please, delete any kind of comment about “higher” or “lower” prevalence, as the total number of samples is quite low, and no statistical evaluation has been carried out.

Lines 235-241: please, delete any kind of comment about “higher” or “lower” prevalence, as the total number of samples is quite low, and no statistical evaluation has been carried out.

Lines 243-258: please, delete any kind of comment about “higher” or “lower” prevalence in “healthy vs unhealthy” pigs, as the inclusion criteria of “healthy/unhealthy” pigs sound too vague, the total number of samples is quite low, and no statistical evaluation has been carried out.

Accordingly, please modify conclusions.

Comments on the Quality of English Language

None

Author Response

Reviewer 3

Comments and Suggestions for Authors

Major comments

Line 61: please, add a suitable reference at this point.

Answer: Done as suggested.

Line 75: please, correct 2000 (maybe, 2020).

Answer: Done as suggested

Lines 80-86: please, add more details about tissue samples, as well as about healthy samples. Moreover, please add details about ethical approval of the present study.

Answer: We have modified the nomenclature to describe the groups, samples are defined as follow: samples of piglets were classified as “growth retardation” or “asymptomatic”, while sows were classified as “reproductive failure” or “asymptomatic”.

Lines 87-94: please, report such data as a table.

Answer: Table 1 was added as suggested.

Lines 95-98: please, add more details about such samples. How was tested/detected the absence of PCV3 genome and anti-PCV3 antibodies? Who is Dr. Gimenez-Lirola? Please, add details about his affiliation. Were such samples from an ethically approved investigation?

Answer: Serum samples were collected from previous study performed by Dr. Gimenez-Lirola’s research group at Iowa State University (Veterinary Diagnostic and Production Animal Medicine, College of Veterinary Medicine, Ames, Iowa). The study was approved by AMVC WeSearch DBA VRI (Audubon, IA, USA) Animal Use and Care Committee (BI-S-18-1248).

Cesarean-derived and colostrum-deprived (CDCD) piglets lack maternal antibodies and are specifically raised as pathogen free animals.  Therefore, the sera from these pigs are free of PCV3 genome and anti-PCV3 antibodies.

Line 150: please, add such data within the materials and methods section.

Answer: The information has been added to a new table as suggested (Table 1).

Lines 160-161: actually, four pools 2015 are reported in table n. 2. Please, check such results.

Answer: The data have been corrected. We had a format problem in table 2.

Lines 162-163: two pools 2021 are reported in table n. 2. Please, check such results.

Answer: The data have been corrected.  We had a format problem in table 2.

Lines 179-185: please, move such data to the materials and methods section.

Answer: Lines 185-188 describe the results of ROC curve, and lines 182-185 provide a brief description to the reader for a better understanding. In consequence, we think the above is important in this section.

Lines 219-220: please, delete any kind of comment about “higher” or “lower” prevalence, as the total number of samples is quite low, and no statistical evaluation has been carried out.

Answer: We remove the comment “higher” as suggested.

Lines 235-241: please, delete any kind of comment about “higher” or “lower” prevalence, as the total number of samples is quite low, and no statistical evaluation has been carried out.

Answer: We performed a statistical analysis to support the comment “higher” or “lower”.

Lines 243-258: please, delete any kind of comment about “higher” or “lower” prevalence in “healthy vs unhealthy” pigs, as the inclusion criteria of “healthy/unhealthy” pigs sound too vague, the total number of samples is quite low, and no statistical evaluation has been carried out.

Accordingly, please modify conclusions.

Answer: We performed a statistical analysis to support the comment “higher” or “lower” for antibody analysis. For lines 219-220, we removed the comment “higher” or “lower” as suggested.

Round 2

Reviewer 2 Report

Comments and Suggestions for Authors

The Authors have corrected most of the critical points.

One minor editing/printing error to be solved: The bar at the phylogenetic tree seems quite strange, please set it.

Author Response

The phylogenetic tree was corrected accordingly.

Reviewer 3 Report

Comments and Suggestions for Authors

None

Comments on the Quality of English Language

None